# Application of a Two-Dimensional Mapping-Based Visualization Technique: Nutrient-Value-Based Food Grouping

**DOI:** 10.3390/nu15235006

**Published:** 2023-12-04

**Authors:** Ryota Wakayama, Satoshi Takasugi, Keiko Honda, Shigehiko Kanaya

**Affiliations:** 1Meiji Co., Ltd., 2-2-1 Kyobashi, Chuo-ku 104-9306, Tokyo, Japan; satoshi.takasugi@meiji.com; 2Computational Systems Biology Laboratory, Division of Information Science, Graduate School of Science and Technology & Data Science Center, Nara Institute of Science and Technology, 8916-5 Takayama-cho, Ikoma 630-0192, Nara, Japan; 3Medicine Nutrition, Faculty of Nutrition, Kagawa Nutrition University, 3-9-21 Chiyoda, Sakado 350-0288, Saitama, Japan

**Keywords:** food classification, machine learning, t-distributed stochastic neighbor embedding, Asia, Japan nutrition, Japanese diets, processed food, food quality, profiling, information science

## Abstract

Worldwide, several food-based dietary guidelines, with diverse food-grouping methods in various countries, have been developed to maintain and promote public health. However, standardized international food-grouping methods are scarce. In this study, we used two-dimensional mapping to classify foods based on their nutrient composition. The Standard Tables of Food Composition in Japan were used for mapping with a novel technique—t-distributed stochastic neighbor embedding—to visualize high-dimensional data. The mapping results showed that most foods formed food group-based clusters in the Standard Tables of Food Composition in Japan. However, the beverages did not form large clusters and demonstrated scattered distribution on the map. Green tea, black tea, and coffee are located within or near the vegetable cluster whereas cocoa is near the pulse cluster. These results were ensured by the k-nearest neighbors. Thus, beverages made from natural materials can be categorized based on their origin. Visualization of food composition could enable an enhanced comprehensive understanding of the nutrients in foods, which could lead to novel aspects of nutrient-value-based food classifications.

## 1. Introduction

To maintain and promote public health, it is important to follow a healthy and balanced diet that has adequate nutrient levels. To encourage the adoption of balanced diets containing diverse foods, the World Health Organization (WHO) established a “Healthy diet” [1], and several food-based dietary guidelines (FBDGs) were developed worldwide. However, food-classification methods for the FBDGs are diverse and differ among countries [2]. In particular, there is a wide variability in nuts, legumes, and animal food sources. For example, legumes are categorized as protein foods, starchy staples, vegetables, and legumes. Nuts are categorized as protein-rich foods, including legumes and fats/oils. In some countries, especially Latin America and the Caribbean, animal-source foods include dairy; however, in most other countries, dairy is considered a separate food group. In Japan, an FBDG called “The Japanese Food Guide Spinning Top” was developed to promote the health policy [3]. The guideline recommends a balanced diet composed of grain dishes (rice, bread, noodles, and pasta), vegetable-based dishes (vegetables, mushrooms, potatoes, and algae), fish and meat dishes (meat, fish, eggs, and soybeans), fruits, and dairy (milk and milk products). These categories are based on Japanese dietary habits [4].

Detailed information on food composition is a prerequisite for implementing dietary guidelines [5]. In Japan, the Standard Tables of Food Composition were developed based on the idea that it is important to clarify the nutrient component of daily intake [6,7]. Since the release of the first edition in 1950, the Standard Tables of Food Composition in Japan have played a role in providing basic data on food components and have been widely used in various fields, such as research and food education. The Eighth Revised Version of the Standard Tables of Food Composition in Japan was published in 2020. These food-composition tables provide insight into the changes in Japanese gastronomic culture and eating habits [7,8] and include the following food groups: cereals, potatoes and starches; sugars and sweeteners; pulses, nuts and seeds; vegetables; fruits; mushrooms; algae; fish, mollusks and crustaceans; meat; eggs, milk and milk products; fats and oils; confectionaries; beverages; seasonings and spices; and prepared foods. In these food-composition tables, foods are categorized by considering Japanese dietary habits and convenience [9]. Therefore, the nutrient-value-based food categorization could improve our understanding of the characteristics of each food group.

The visualization of high-dimensional datasets, such as food-composition tables, increases awareness of the nutritional value of foods. Data-reduction techniques are useful for investigating high-dimensional datasets. Principal component analysis (PCA) is one of the techniques [10] that have been applied to food-composition tables [11,12,13]. Recently, a new technique called the t-distributed stochastic neighbor embedding (t-SNE) was developed for visualizing high-dimensional data [14]. A previous study showed that t-SNE performs better than PCA in visualizing a food-composition table [15]. However, food classification based on nutrient values has not yet been performed using the t-SNE method.

This study aimed to classify foods based on nutrient information in the Standard Tables of Food Composition in Japan, 2020 (Eighth Revised Edition). We used the t-SNE mapping method and the k-nearest neighbors (k-NN), which lead to the estimation of appropriate nutrient intake through FBDGs. This nutrient value-based evaluation of food would enable a comprehensive understanding of the relationship between nutrients and food.

## 2. Materials and Methods

### 2.1. Standard Tables of Food Composition in Japan, 2020 (Eighth Revised Edition)

The Standard Tables of Food Composition in Japan, 2020 (Eighth Revised Edition) is an open source publication from the Ministry of Education, Culture, Sports, Science and Technology, Japan, in which 2478 foods were divided into 17 food groups. In this study, 16 food groups, except for “Prepared foods”, were used for analysis; “Prepared foods” was included in meal. The 16 food groups and number of foods in each group are shown in Table 1. The standard tables comprise 54 nutrients.

### 2.2. Data Imputation

In Table 2, we cannot ignore the missing values of nutrients for foods, that is, missing values exist in 26–27% of foods for saturated fatty acids, *n*-6 PUFA and *n*-3 PUFA, and 1–8% of foods for vitamins D, E, K, B_6_, B_12_ and C; folic acid, pantothenic acid; zinc, copper, and manganese. Therefore, the imputation method was applied to estimate missing values using multidimensional scaling (MDS), which represents measurements of similarity among pairs of objects as distances between points of a low-dimensional multidimensional space, and random forest regression methods based on four steps (Figure 1).

Step 1: The distance between objects can be calculated using Equation (1).
(1)$$d\left(i,i’\right)=\frac{M}{U}d_{i,i’}\left(U\right)$$
where $d_{i,i’}\left(U\right)$ represents the distance between variables without missing values in the two objects and *M* is the total number of variables. Thus, the distance matrix *D* was calculated from the original dataset *X.* Step 2: MDS variables *U* for all objects based on the distance matrix *D* calculated by Equation (1) can be obtained based on the MDS. Here *U* has compressed information of *X*. Step 3: A regression model of *j*th variable of *x*_j_ without missing values based on *U* was created by random forest. Step 4: Using the random forest regression models, missing values were calculated for *j*th variable. Steps 2 and 3 were performed for all variables *X*.

### 2.3. Missing Values

The accuracy of the missing values was validated by removing 50 known values and estimating the values for the imputation method as described in the Methods Section (Section 2.2). The average correlation coefficient for the 30 variables was 0.984 (range 0.957–0.997). Thus, the imputation values for missing values are expected to be highly accurate. Furthermore, as shown in Table 1, the Standard Tables of Food Composition in Japan, 2020 (Eighth Revised Edition) include 2428 entries in 16 food groups; however, these are not the complete data. Data imputation was performed to increase the amount of data available for mapping. In addition to energy and water, nutrients that had an estimated average requirement (EAR), recommended dietary allowance (RDA), adequate intake (AI), or a tentative dietary goal for preventing lifestyle-related diseases (DG) in the Japanese Dietary Reference Intakes [16,17] were chosen. However, nutrients with approximately 50% of the total data were excluded (Table 2). EAR is composed of proteins, vitamin A, vitamin B_1_, vitamin B_2_, niacin equivalent, vitamin B_6_, vitamin B_12_, folic acid, vitamin C, sodium, calcium, magnesium, iron, zinc, copper, iodine, selenium, and molybdenum. RDA is composed of proteins, vitamins A, B_1_, B_2_, niacin equivalents, B_6_, B_12_, folic acid, C, calcium, magnesium, iron, zinc, copper, iodine, selenium, and molybdenum. AI is composed of *n*-6 polyunsaturated fatty acids, *n*-3 polyunsaturated fatty acids, vitamin D, vitamin E, vitamin K, pantothenic acid, biotin, potassium, phosphorus, manganese, and chromium. DG is composed of proteins, fats, saturated fatty acids, carbohydrates, dietary fiber, sodium, and potassium. In this case, the selected nutrients were energy, water, protein, fat, saturated fatty acid, *n*-3 polyunsaturated fatty acid, *n*-6 polyunsaturated fatty acid, dietary fiber, carbohydrate, sodium, potassium, calcium, magnesium, phosphorus, iron, zinc, copper, manganese, vitamin A (retinol activity equivalents), vitamin D, vitamin E (alpha-tocopherol), vitamin K, vitamin B_1_, vitamin B_2_, niacin equivalent, vitamin B_6_, vitamin B_12_, folic acid, pantothenic acid, and vitamin C, whereas iodine, selenium, chromium, molybdenum, and biotin were excluded.

### 2.4. Mapping

Nutrient-density data were used for mapping to exclude the effect of water. Thus, the data for nutrients per 100 g of edible portion were converted to per 100 kcal. Water and energy data were excluded from the dataset during conversion. All nutrients were mapped. Furthermore, to evaluate the contribution of macronutrients (proteins, fats, and carbohydrates) to the mapping, we performed mapping using nutrients without them.

A two-dimensional map of the nutrient matrix consisting of nutrients per 100 kcal and 2221 foods was constructed using t-distributed stochastic neighbor embedding (t-SNE). t-SNE is a nonlinear dimensionality reduction method in which Gaussian probability distributions over a high-dimensional space are constructed and used to optimize a Student’s t-distribution in a low-dimensional space. The low-dimensional embedding descriptors, that is, nutrients per 100 kcal, can be obtained by minimizing the Kullback–Leibler divergence [18] between the distributions in high- and low-dimensional spaces, maintaining pair-wise similarity to the high-dimensional space. Therefore, it is a powerful tool for understanding the relationship among objects as a low-dimensional space because the objects close to each other on the low-dimensional surface represent states that are similar in a high-dimensional space [14,19]. The performance of t-SNE is robust with sufficiently large hyper parameters [14,19]. Mapping was performed using the R Package (version 4.2.2).

### 2.5. Food Classification

The k-NN, a type of machine-learning model [20,21], was applied to classify foods based on nutrient information (k = 3). Nutrient data per 100 kcal were used for food classification and mapping. The misclassification rate was calculated by dividing the number of incorrect predictions by the number of the total predictions.

## 3. Results

All data in the Standard Tables of Food Composition in Japan, 2020 (Eighth Revised Edition) were imputed. The numbers of foods consumed before and after data imputation are presented in Table 1. The original food-composition table included 2428 foods in 17 food groups; however, the data for only 1619 foods were analyzable without any missing values. After data imputation, the number of foods that contained all nutrients was 2221 (/100 kcal). The accuracy of the missing values was validated by removing 50 known values and estimating the values by using the imputation method described in the Methods Section (Section 2.3). The correlation coefficients for the 30 variables averaged 0.984 (range 0.957–0.997), which is very strong [22]. Thus, the imputation values for missing values are expected to be highly accurate. The accuracy of data imputation is presented in Appendix A.

Two-dimensional mapping was performed using/100 kcal data after data imputation. The results of mapping all nutrients using t-SNE are shown in Figure 2. Figure 3 shows the effects of each nutrient (Figure 2). Macronutrients (proteins, fats, and carbohydrates) were distributed in three directions: proteins were located in the first quadrant, fat in the second quadrant, and carbohydrates in the third quadrant. In contrast, other nutrients were located near the center of the map. Phosphorus, zinc, vitamin D, vitamin B_1_, niacin equivalents, vitamin B_12_, and pantothenic acid were present in the first quadrant. Moreover, *n*-3 polyunsaturated fatty acids, *n*-6 polyunsaturated fatty acids, dietary fiber, sodium, potassium, calcium, magnesium, iron, copper, manganese, vitamins A, D, E, K, B_2_, B_6_, folic acid, and biotin were located in the fourth quadrant.

The k-NN method was used for ensuring classification accuracy. The number of foods misclassified using k-NN is shown in Table 3, and the average misclassification rate was 13%.

The results of the mapping with nutrients, excluding macronutrients (proteins, fats, and carbohydrates), are shown in Figure 4. Furthermore, Figure 5 shows the effects of each nutrient (Figure 4). *n*-6 polyunsaturated fatty acids, dietary fiber, manganese, and vitamin C are located in the first quadrant. Sodium is present in the second quadrant. *n*-3 polyunsaturated fatty acids, magnesium, phosphorus, iron, zinc, copper, vitamin D, vitamin B2, niacin equivalents, vitamin B6, vitamin B12, and pantothenic acid is located in the third quadrant. Saturated fatty acids, potassium, calcium, vitamins A, E, K, B1, and folic acid is located in the fourth quadrant.

## 4. Discussion

### 4.1. Discussion

This is the first visualization study which was undertaken using t-SNE with the Standard Tables of Food Composition in Japan, 2020 (Eighth Revised Edition). Furthermore, this study showed that visualization by two-dimensional mapping was possible based on nutrient information in food-composition tables, as shown in a previous study [15]. Previous research using PCA has shown that nuts, seeds, and pulses were well separated; however, fruits and vegetables were not separated adequately [13]. However, the results of this study showed that these foods were well-separated. In addition, as shown in a previous study that indicated that t-SNE was better than PCA for the visualization of food tables [15], this study using t-SNE showed that it was possible to classify many foods into distinct food groups by focusing on the nutrients in the dietary reference intake for the Japanese population.

The results showed that most foods formed clusters in accordance with the food groups in the Standard Tables of Food Composition in Japan, 2020 (Eighth Revised Edition), as indicated by the misclassification rate. This indicates that many food groups had common nutrient patterns and were categorized based on nutrient similarity. Furthermore, as shown in Figure 4, which was performed without protein, fat, or carbohydrates, many foods formed clusters in accordance with the food groups. This indicates that the characteristics of the food groups were determined by micro- and macronutrient (protein, fat, and carbohydrate) balance. Given that the results of this study are based on nutrient density, nutrient density may also contribute to the classification of foods.

In contrast, the beverages did not form large clusters and were scattered. The misclassification rate of beverages was 48%, with the highest rate among the food groups. This may be because this food group comprises a wide range of beverages, including soft drinks, beverages made from natural materials (green tea, black tea, kelp tea, and cocoa), and alcoholic drinks. In Figure 2, the coordinates of green teas (refined, powdered, and middle grade) and black tea were as follows: (26.8, −19.6), (27.7, −14.3), (27.0, −18.8), and (25.2, −18.5), which fell within the vegetables cluster. The k-NN results showed that green tea (refined and powdered) and black tea could be classified as vegetables. Green tea consumption has been reported to be beneficial to overall health [23]. Generally, tea is composed of the leaves and buds of Camellia sinensis and the difference between green tea and black tea lies in the manufacturing process [24]. In addition to tea, the coordinates of coffee and instant coffee were (5.5, −19.7) and (9.7, −15.6), which were close to the vegetables cluster. The coordinates of kelp tea, which was categorized in beverages, were (−2.5, −10.8), which were near the processed kelps (salted kelp and kelp boiled in soy sauce). In addition, the coordinates of cocoa (pure powder), an ingredient of dark chocolate [25], were (−3.6, 5.8), which were close to the pulses cluster. The food was classified into pulses using k-NN. Cocoa is rich in polyphenols and has beneficial effects [26,27,28]. Moreover, cocoa contains nitrogenous proteins and minerals, such as potassium, magnesium, phosphorus, iron, zinc, and copper [29]. Furthermore, pulses are good sources of polyphenols [30], proteins, and minerals, such as magnesium, iron, and zinc [31]. Although this study did not consider polyphenols, there are many common features between cocoa and pulses, as shown in Figure 2 in terms of those micronutrients. These results indicate that the original materials were reflected in the mapping results.

There is no international consensus on the definition of vegetables and fruits; thus, definitions vary among countries or cultures [32]. The Standard Tables of Food Composition in Japan, 2020 (Eighth Revised Edition) categorized vegetables of herbaceous plants, such as banana, strawberry, melon, watermelon, and pineapple, into fruits because of their convenience and Japanese dietary habits. From the results of Figure 2, their coordinates were (6.2, −38.5), (7.9, −33.3), (6.7, −37.5), (5.5, −36.2), and (5.3, −37.2), respectively, and they were all within the fruits cluster. These results indicate that mapping distinguishes vegetables from fruits in terms of nutrients.

Figure 3 shows that the mapping shown in Figure 2 is largely affected by proteins, fats, and carbohydrates. This means that foods rich in protein are drawn in the first quadrant, foods rich in fat are drawn in the second quadrant, foods rich in carbohydrates are drawn in the third quadrant, and foods rich in micronutrients are drawn between the first and fourth quadrants. These results indicate that the intake of a variety of foods was associated with the intake of a variety of nutrients. It is important to consume a diet containing a variety of foods. For example, the Dietary Approaches to Stop Hypertension (DASH) diet is composed of rich fruits, vegetables, and low-fat dairy foods, which have significant effects on blood pressure [33,34]. In addition to a DASH diet, a Mediterranean diet rich in olive oil, fruits, nuts, vegetables, and cereals, moderate in fish, and poor in red and processed meats was associated with low cardiovascular risk [35]. The WHO recommends a diversified and balanced diet and intake of foods such as fruits, vegetables, nuts, and pulses to protect against malnutrition and noncommunicable diseases [1]. For Japanese people, consuming diverse foods and increasing the amount of fruit and pulses are inversely associated with mortality [36]. Based on the results of this study, these foods are located in areas where various micronutrients are found.

The Japanese Food Guide Spinning Top was developed by the Ministry of Health, Labour, and Welfare and the Ministry of Agriculture, Forestry, and Fisheries of Japan in 2005. A diet that adheres to the Japanese Food Guide Spinning Top decreases the risk of total mortality and mortality from cardiovascular disease [37]. Figure 2 shows that the foods categorized in each dish were situated close to each other. This indicates that the mapping of food-composition tables can explain the relationship between the Japanese Food Guide Spinning Top and nutrient-based foods.

### 4.2. Limitation

This study has some limitations. First, information on iodine, selenium, chromium, molybdenum, and biotin was not added to the dataset for data imputation and mapping because the number of available foods was insufficient. Secondly, details of bioactive compounds, such as phytochemicals and peptides, and the quality of nutrients, such as amino acid balance and fatty acid composition, were not included in the dataset of this study. Finally, this study employed nutrient density and did not reflect the actual nutrient quantity of real intake in the mapping. When data from the Standard Tables of Food Composition in Japan increase and nutrient data become available, more precise mapping will be performed. Analysis of the quality of nutrients or other compounds may lead to a better understanding of food characteristics.

## 5. Conclusions

In conclusion, this is the first report on t-SNE-based visualization of the Standard Tables of Food Composition in Japan, 2020 (Eighth Revised Edition), and the results showed that most foods formed clusters in accordance with the food groups in the Standard Tables of Food Composition in Japan, 2020 (Eighth Revised Edition) in terms of nutrient values. In contrast, beverages did not form a large cluster, and beverages made from natural materials such as green tea, black tea, coffee, or cocoa were categorized based on their origin. The visualization of food composition enables a comprehensive understanding of the nutrients in foods, thereby leading to novel aspects of food classifications based on nutrient values.

## Figures and Tables

**Figure 1 nutrients-15-05006-f001:**
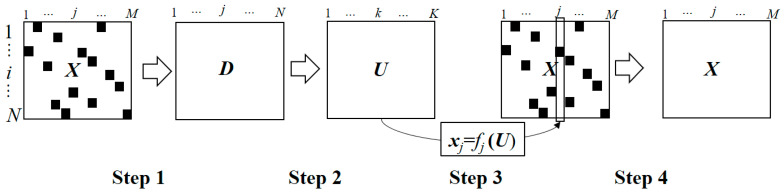
Imputation with MDS and random forest with 4 steps. Original matrix and distance matrix are determined by the numbers of foods (*N*) and nutrient value/kcal (*M*). In MDS matrix *U*, we set *K* as *M* in *i* and *i*’th objects, respectively.

**Figure 2 nutrients-15-05006-f002:**
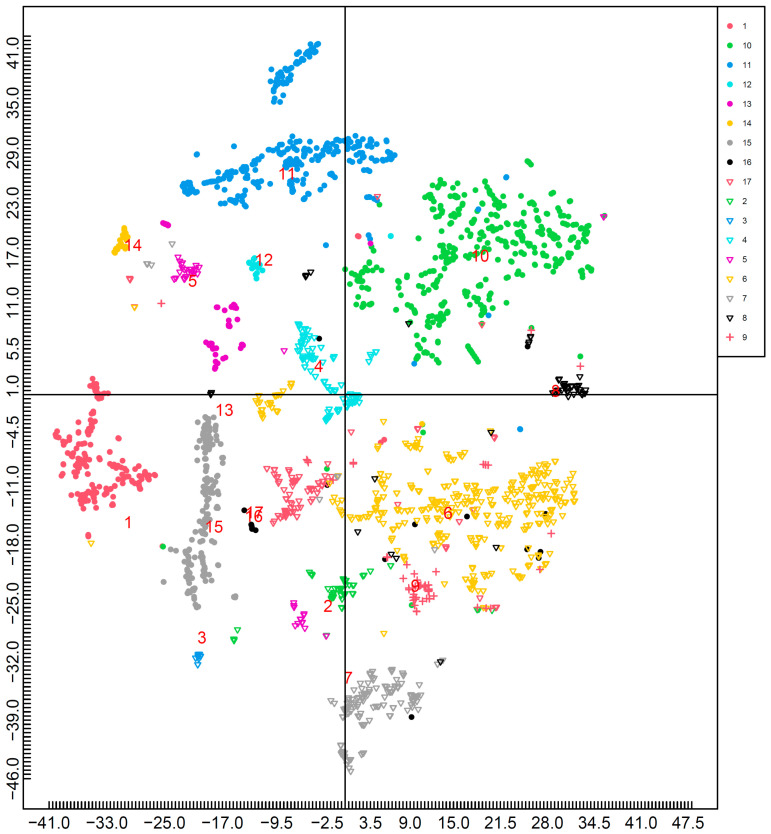
Mapping by t-SNE (with all nutrients). This figure shows the mapping based on all nutrients. The locations indicated by red numbers are the median of each food group: 1: cereals; 2: potatoes and starches; 3: sugars and sweeteners; 4: pulses; 5: nuts and seeds; 6: vegetables; 7: fruits; 8: mushrooms; 9: algae; 10: fish, mollusks, and crustaceans; 11: meat; 12: eggs; 13: milk and milk products; 14: fats and oils; 15: confectionaries; 16: beverages; and 17: seasonings and spices.

**Figure 3 nutrients-15-05006-f003:**
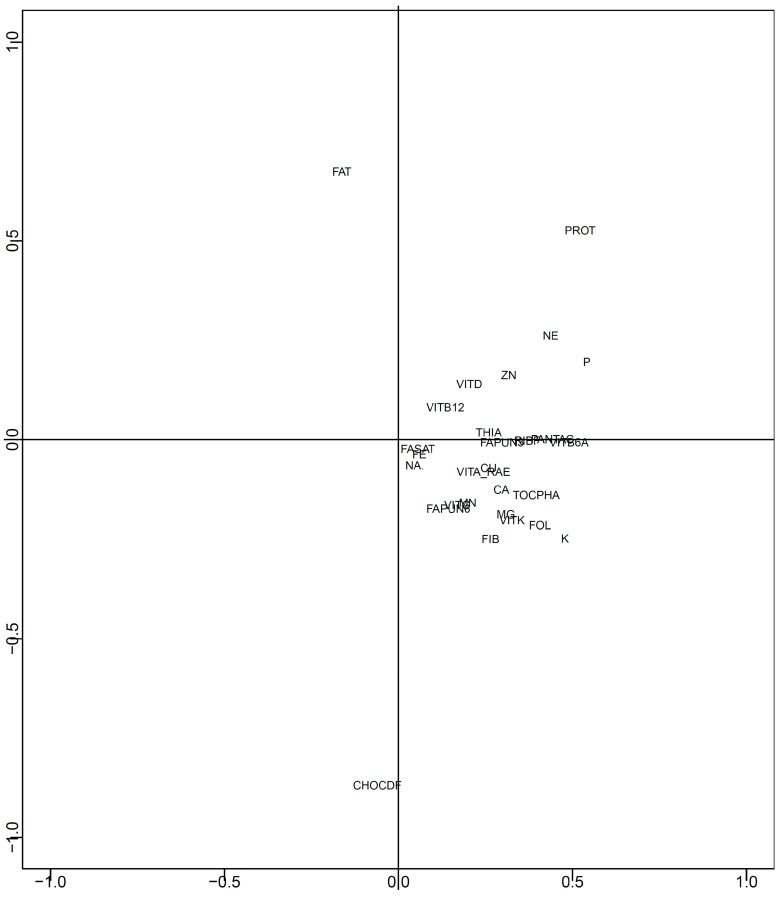
Effects of each nutrient in Figure 2 (with all nutrients). Effects of each nutrient in Figure 2 are shown. PROT: protein, FAT: fat, FASAT: saturated fatty acid, FAPUN3: *n*-3 polyunsaturated fatty acid, FAPUN6: *n*-6 polyunsaturated fatty acid, FIB: dietary fiber, CHOCDF: carbohydrate, NA: sodium, K: potassium, CA: calcium, MG: magnesium, P: phosphorus, FE: iron, ZN: zinc, CU: copper, MN: manganese, VITA_RAE: vitamin A (retinol activity equivalents), VITD: vitamin D, TOCPHA: vitamin E (alpha-tocopherol), VITK: vitamin K, THIA: vitamin B1, RIBF: vitamin B2, NE: niacin equivalent, VITB6A: vitamin B6, VITB12: vitamin B12, FOL: folic acid, PANTAC: pantothenic acid, VITC: vitamin C.

**Figure 4 nutrients-15-05006-f004:**
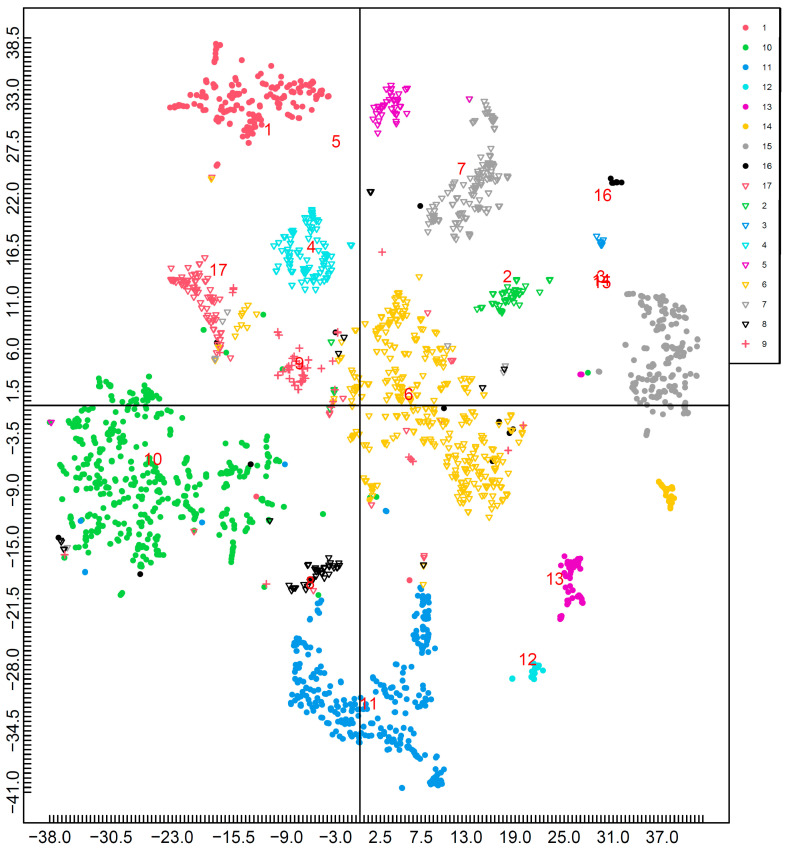
Mapping by t-SNE (without PFC). This figure shows mapping based on the nutrients without PFC. The location shown by red numbers is the median of each food group: 1: cereals; 2: potatoes and starches; 3: sugars and sweeteners; 4: pulses; 5: nuts and seeds; 6: vegetables; 7: fruits; 8: mushrooms; 9: algae; 10: fish, mollusks, and crustaceans; 11: meat; 12: eggs; 13: milk and milk products; 14: fats and oils; 15: confectionaries; 16: beverages; and 17: seasonings and spices.

**Figure 5 nutrients-15-05006-f005:**
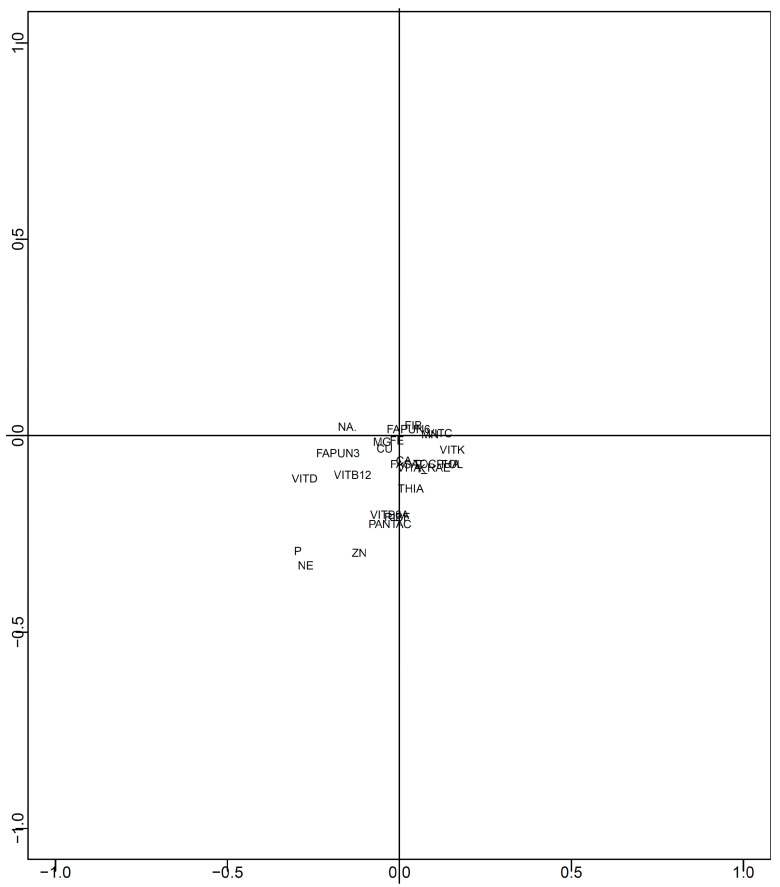
Effects of each nutrient in Figure 4 (without PFC). Effects of each nutrient in Figure 4 are shown. FASAT: saturated fatty acid, FAPUN3: *n*-3 polyunsaturated fatty acid, FAPUN6: *n*-6 polyunsaturated fatty acid, FIB: dietary fiber, NA: sodium, K: potassium, CA: calcium, MG: magnesium, P: phosphorus, FE: iron, ZN: zinc, CU: copper, MN: manganese, VITA_RAE: vitamin A (retinol activity equivalents), VITD: vitamin D, TOCPHA: vitamin E (alpha-tocopherol), VITK: vitamin K, THIA: vitamin B1, RIBF: vitamin B2, NE: niacin equivalent, VITB6A: vitamin B6, VITB12: vitamin B12, FOL: folic acid, PANTAC: pantothenic acid, VITC: vitamin C.

**Table 1 nutrients-15-05006-t001:** Number of foods in food groups.

Food Group	Number of Foods		
	Before Data Imputation		After Data Imputation
	All Data	Complete Values	/100 kcal
1 Cereals	205	145	189
2 Potatoes and starches	70	37	60
3 Sugars and sweeteners	30	0	17
4 Pulses	108	74	105
5 Nuts and seeds	46	38	46
6 Vegetables	401	246	381
7 Fruits	183	79	145
8 Mushrooms	55	44	55
9 Algae	57	39	56
10 Fish, mollusks, and crustaceans	453	377	453
11 Meat	310	289	310
12 Eggs	23	22	23
13 Milk and milk products	59	43	58
14 Fats and oils	34	28	31
15 Confectionaries	185	105	176
16 Beverages	61	12	23
17 Seasonings and spices	148	41	93
Total	2428	1619	2221

This table shows the number of foods in the food group before and after data imputation. The original food tables contained 2428 foods (“all data”) and 1619 foods without missing values (“complete values”). After data imputation, values of 2221 foods per 100 kcal were available. “All data” means the number of foods in the original food tables and “complete values” means the number of foods in the original food without missing values.

**Table 2 nutrients-15-05006-t002:** Number of foods in each nutrient group (the number of foods and percentage of total foods reported in each nutrient of the Standard Tables of Food Composition in Japan 2020).

Nutrient/Energy			Number of Foods	Percentage of Missing Values (%)	Selected in This Work
Energy			2429	0	✓
Water			2429	0	✓
Protein			2427	0	✓
Dietary Fats		Fat	2427	0	✓
		Saturated fatty acid	1787	26	✓
		*n*-6 PUFA	1777	27	✓
		*n*-3 PUFA	1783	27	✓
Carbohydrates		Carbohydrate	2428	0	✓
		Dietary fiber	2321	4	✓
Vitamins	Fat-soluble	Vitamin A	2405	1	✓
		Vitamin D	2340	4	✓
		Vitamin E	2320	4	✓
		Vitamin K	2242	8	✓
	Water-Soluble	Vitamin B1	2421	0	✓
		Vitamin B2	2424	0	✓
		Niacin equivalent	2428	0	✓
		Vitamin B6	2387	2	✓
		Vitamin B12	2341	4	✓
		Folic acid	2402	1	✓
		Pantothenic acid	2390	2	✓
		Biotin	1242	49	
		Vitamin C	2398	1	✓
Minerals	Macro	Sodium	2425	0	✓
		Potassium	2426	0	✓
		Calcium	2426	0	✓
		Magnesium	2418	0	✓
		Phosphorus	2426	0	✓
	Micro	Iron	2426	0	✓
		Zinc	2415	1	✓
		Copper	2415	1	✓
		Manganese	2280	6	✓
		Iodine	1238	49	
		Selenium	1249	49	
		Chromium	1248	49	
		Molybdenum	1238	49	

**Table 3 nutrients-15-05006-t003:** Comparison of the actual and the predicted food groups.

		Predictive Food Group																	MR (%)
		1	2	3	4	5	6	7	8	9	10	11	12	13	14	15	16	17	
Actual food group	1	178	0	0	0	0	0	1	0	0	0	0	0	0	0	10	0	0	6
	2	1	56	0	0	0	0	1	0	0	0	0	0	0	0	2	0	0	7
	3	0	2	13	0	0	0	0	0	0	0	0	0	0	0	2	0	0	24
	4	0	0	0	101	0	1	0	0	0	1	0	0	0	0	1	0	1	4
	5	1	2	0	1	38	1	1	0	0	0	0	0	1	0	1	0	0	17
	6	2	5	0	1	0	366	4	1	0	0	0	0	0	0	0	0	2	4
	7	1	4	1	0	2	3	130	2	0	0	0	0	0	0	0	0	2	10
	8	1	0	0	0	0	6	1	45	0	1	0	0	0	0	1	0	0	18
	9	0	2	0	0	0	6	0	2	46	0	0	0	0	0	0	0	0	18
	10	2	0	0	1	1	1	0	1	1	443	2	0	0	0	0	0	1	2
	11	1	0	0	0	0	0	0	0	0	11	298	0	0	0	0	0	0	4
	12	0	0	0	0	0	0	0	0	0	0	0	22	0	0	1	0	0	4
	13	1	0	0	0	0	0	1	0	0	2	2	0	45	1	6	0	0	22
	14	0	0	0	0	0	0	0	0	0	1	1	0	0	29	0	0	0	6
	15	6	3	1	0	0	0	1	0	0	0	0	0	2	0	163	0	0	7
	16	0	0	0	1	0	6	1	0	0	0	0	0	0	0	1	12	1	48
	17	2	0	0	2	0	7	1	0	0	1	1	1	0	0	1	1	75	19

This table shows the number of foods that were classified correctly or incorrectly using the k-NN method. The food groups included: 1: cereals; 2, potatoes and starches; 3, sugars and sweeteners; 4, pulses; 5, nuts and seeds; 6, vegetables; 7, fruits; 8, mushrooms; 9, algae; 10, fish; mollusks and crustaceans; 11, meat; 12, eggs; 13, milk and milk products; 14, fats and oils; 15confectionaries; 16, beverages; 17, seasonings and spices. MR: the misclassification rate for each food group.

## Data Availability

This data can be found here: https://www.mext.go.jp/a_menu/syokuhinseibun/mext_01110.html (accessed on 30 November 2023).

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
