# Peer review of "Application of a Two-Dimensional Mapping-Based Visualization Technique: Nutrient-Value-Based Food Grouping"

_nutrients, 2023, doi:10.3390/nu15235006_

Round 1

Reviewer 1 Report

Comments and Suggestions for Authors

Interesting and creative study to complemenanalysis of the nutrient density of a diet as may be the case of dietary guidelines proposed for a specific population.

Also us the studies point out, with this metods you can visualize the nutrients gap  in a dietary pattern,wich is I why suggest including in objetives the dimensioning of the quality of the diet through the analysis of the nutrient density of a certain consumption pattern. Isuggest point out that this method complements the grouping of the food table since they are grouped by nutrients density, not only by the macro and micro nutriens components that the table generally present. 

Author Response

Dear Reviewer 1,

Thank you for your insightful comments and suggestions. We appreciate your positive feedback on our study. We agree with your suggestion to include the dimensioning of diet quality through the analysis of nutrient density in our objectives. This will indeed provide a more comprehensive view of the dietary pattern.

We also acknowledge your point about the method complementing the grouping of the food table. We will suggest in our revised manuscript that our method groups foods based on nutrient density, not just the macro and micronutrient components typically presented in food tables.

Once again, thank you for your valuable input. In addition to t-SNE, we also examined supervised machine learning, k-nearest neighbors (kNN). The classification of t-SNE is consistent with that in kNN. Therefore, we think that our results are robust. However, as you pointed out, the contribution of nutrient density could also be considered. Therefore, we will incorporate your suggestions in our revised manuscript (in Line 257) to improve the clarity and applicability of our study. The following sentence will be added to the revised manuscript in Line 257: Given that the results of this study are based on nutrient density, nutrient density may also contribute to the classification of foods.

Reviewer 2 Report

Comments and Suggestions for Authors

Dear Autors,

The manuscript submitted for review is quite interesting, but it does not explain how such visualized data can be practically used, and I do not understand the conclusion that such mapping can improve the consumption of foods from a given group. The article was formatted quite sloppily. Some sentences from the previous pages seem unfinished (pages 2 and 3).

line 34/ line 140 - There's the same Table number. Therefore the authors should be check the numbers of Table 1, table 2 and Table 3 .

Line 177 - Error Reference source not found - should be corrected.

Why is the size of Fig. 5 so large even though there is practically no data in it?

References - positions 6 and 7 - need correction.

The English language in the manuscript is understandable   Reviewer

Author Response

Dear Reviewer 2,

Thank you for your detailed review and constructive feedback. We appreciate your time and effort in reviewing our manuscript. We understand your concerns and agree that there are areas in our manuscript that need improvement. Here’s how we plan to address your comments:

but it does not explain how such visualized data can be practically used, and I do not understand the conclusion that such mapping can improve the consumption of foods from a given group.

Reply: Understanding the nutritional content of food can be challenging for the general public, especially those without a strong background in nutrition. We think that it is conceivable to use the results of mapping in the nutritional education for the general public. However, as you pointed out, this concept is not explored in depth in this study, and we overstepped in our conclusion. Thus, we will remove the following sentence in the conclusion in the revised manuscript: These findings can contribute to appropriate nutrient intake through the FBDGs.

Some sentences from the previous pages seem unfinished (pages 2 and 3).

Reply: We found the unfinished sentences and will correct these errors in pages 2 and 3 in the revised manuscript.

line 34/ line 140 - There's the same Table number. Therefore, the authors should be checking the numbers of Table 1, table 2 and Table 3.

Reply: We will correct the table numbers (Table 1, Table 2, and Table 3) in Lines 131,138, and 205 in the revised manuscript to avoid any confusion. Along with the revision, we will correct the table numbers in Lines 80, 82, 108, 115, 167, and 187 in the revised manuscript.

Line 177 - Error Reference source not found - should be corrected.

Reply: We found the correct reference source and will correct this error in Line 175 in the revised manuscript.

Why is the size of Fig. 5 so large even though there is practically no data in it?

Reply: Figure 5 shows that foods can be classified only with nutrients excluding protein, fat, and carbohydrate. We recognize that Figure 5 is the important data to show the impact of each nutrient in Figure 4.

References - positions 6 and 7 - need correction.

Reply: We will modify the references at positions 6 and 7 in Line 45 and 51 in the revised manuscript.

Reviewer 3 Report

Comments and Suggestions for Authors

In this paper the authors describe the application of a two-dimensional mapping to classify foods by their composition and in Japan's national food composition tables. The mapping results showed that most of the foods formed clusters based on food groups, and that the visualization technique could improve the global understanding of the nutrients present in foods, which could lead to to new aspects of food classifications.

The work is well written and of great interest for the stackeholders  nutritional tables. In any case, I think the document needs some small changes.

line 28: I think the word "follow" is more appropriate than "consume"

table 1: line 134. Change the sentence “without missing values” to “complete values” or similar words. Insert a caption under the figure where you specify the meaning and the elements of the table.

line 137 remove the repetition in parentheses.

line 313: insert a subparagraph dedicated to the limits of the study, as these are not negligible

Author Response

Dear Reviewer 3,

Thank you for your constructive feedback and suggestions. We appreciate the time and effort you’ve put into reviewing our paper. Here’s how we plan to address your comments:

line 28: I think the word "follow" is more appropriate than "consume"

Reply: We agree that “follow” might be more appropriate than “consume” in this context. We will make this change in Line 28 in the revised manuscript.

table 1: line 134. Change the sentence “without missing values” to “complete values” or similar words. Insert a caption under the figure where you specify the meaning and the elements of the table.

Reply: We will change the phrase “without missing values” to “complete values” in Table 1 and Line 134 in the revised manuscript and add the following sentence under the figure to specify the meaning and the elements of the table in Line 135 in the revised manuscript: “All data” means the number of foods in the original food tables and “complete values” means the number of foods in the original food without missing values.

line 137 remove the repetition in parentheses.

Reply: We will remove the repetition in parentheses in Line134 in the revised manuscript.

line 313: insert a subparagraph dedicated to the limits of the study, as these are not negligible

Reply: We will insert a subparagraph dedicated to the limits of the study in Lines 238 and 314 in the revised manuscript.

We hope these revisions will address your concerns and improve the quality of our manuscript. Once again, thank you for your valuable input.